# An Indoor-Monitoring LiDAR Sensor for Patients with Alzheimer Disease Residing in Long-Term Care Facilities

**DOI:** 10.3390/s22207934

**Published:** 2022-10-18

**Authors:** Ji-Eun Joo, Yu Hu, Sujin Kim, Hyunji Kim, Sunyoung Park, Ji-Hoon Kim, Younghyun Kim, Sung-Min Park

**Affiliations:** 1Department of Electronic and Electrical Engineering, Ewha Womans University, Seoul 03760, Korea; 2Graduate Program in Smart Factory, Ewha Womans University, Seoul 03760, Korea; 3Department of Electronic and Computer Engineering, University of Wisconsin–Madison, Madison, WI 53706, USA

**Keywords:** Alzheimer, APD, CMOS, LiDAR, LTCF, NPU, optoelectronic

## Abstract

This paper introduces an indoor-monitoring LiDAR sensor for patients with Alzheimer disease residing in long-term care facilities (LTCFs), and this sensor exploits an optoelectronic analog front-end (AFE) to detect light signals from targets by utilizing on-chip avalanche photodiodes (APDs) realized in a 180 nm CMOS process and a neural processing unit (NPU) used for motion detection and decisions, especially for incidents of falls occurring in LTCFs. The AFE consists of an on-chip CMOS P^+^/N-well APD, a linear-mode transimpedance amplifier, a post-amplifier, and a time-to-digital converter, whereas the NPU exploits network sparsity and approximate processing elements for low-power operation. This work provides a potential solution of low-cost, low-power, indoor-monitoring LiDAR sensors for patients with Alzheimer disease in LTCFs.

## 1. Introduction

Early fall detection is an essential aspect of providing the necessary medical response in a timely manner to older patients with Alzheimer disease or senile dementia residing in long-term care facilities (LTCFs) or homes [1,2,3]. With the advancement of sensing and signal processing technologies, various fall detection methods have been proposed. Using a wearable device is a common approach [4], but this requires patients to manage and wear an extra device, which can be bothersome to those who have physical conditions. However, image-based fall detection has been gaining popularity due to the emergence of artificial intelligence (AI)-based high-accuracy image recognition and its easy applicability without involving patients. However, high accuracy and the convenience of image-based fall detection come at the cost of increased privacy concerns. The ability to visually monitor patients 24 h a day makes them reluctant to adopt it despite its healthcare benefits. In addition, AI-based image recognition often requires the high processing power of computing clouds. Sharing personal images with third parties for AI processing raises even more privacy concerns [5]. Without addressing the privacy concerns, patients will be reluctant to adopt such monitoring systems, which may lead to health-related consequences. To address this problem, researchers have investigated the use of non-traditional imaging technologies, such as mmWave [6] and ultra-wideband [7], for privacy-preserving monitoring. Similarly, a solution for a privacy-preserving yet accurate AI-based fall detection technique is required.

Recently, light detection and ranging (LiDAR) sensors have received a great deal of attention in various fields, such as navigation systems for robots, indoor mapping on mobile devices, and in-home patient monitoring. Because LiDAR sensors generate images with depth information but without detailed red, green, and blue (RGB) information that differentiates individuals, it is considered more suitable for privacy-sensitive applications [8,9]. Figure 1 shows examples of 2D RGB images [10] and the 3D depth images generated from the 2D RGB images by a neural network [11]. As shown in these examples, 3D depth images can be utilized for detecting the posture of objects, but it does not reveal their identities.

In this paper, we propose a design for a LiDAR sensor and a neural processing unit (NPU) that exploit privacy-preserving fall detection at edges. Figure 2 illustrates the feasible usage of the proposed indoor-monitoring LiDAR sensor, which includes a single-chip LiDAR analog front-end (AFE) integrated circuit and an NPU, equipped in the living room of an LTCF. Currently, most LiDAR sensors exploit the principle of the pulsed time-of-flight (ToF) mechanism so that light pulses can be emitted from a transmitter to targets located within a feasible range, and their reflected signals can be detected by an optical receiver. By knowing the speed of light, the target distance can be measured using the time interval between the transmitted (aka START) pulse and its reflected (aka STOP) one. Even for indoor home-monitoring applications, the dynamic range of the received pulses (or echoes) should be wide enough, i.e., greater than 1:1000, thus providing a fast response.

Figure 3a depicts a block diagram of the proposed indoor-monitoring LiDAR sensor, in which the reflected light pulses are detected by an optical detector (typically an avalanche photodiode or APD) in the receiver. Then, the corresponding photocurrents are generated from the APD, converted to voltage signals, and amplified by an analog front-end (AFE) circuit that comprises a transimpedance amplifier (TIA) and a post-amplifier (PA). Thereafter, a time-to-digital converter (TDC) estimates the distance to targets by measuring the time interval between the emitted pulse and the reflected one.

A digital signal processor, i.e., the neural processing unit (NPU) in this work, is used for motion detection and decisions in cases of falling incidents occurring in LTCFs. Figure 3b shows a block diagram of the proposed low-power NPU, which conducts feature extraction for motion detection. In particular, a low-power NPU is required because the indoor-monitoring LiDAR sensor should always be turned on, even with a limited battery, hence providing a data-sparsity-aware NPU with approximate processing elements. This architecture includes a dedicated hardware accelerator for computing layers. Moreover, the timing delay should be effectively reduced to instantly detect fall incidents. By reducing parameters such as weights and activations, it is possible to reduce the amount of computation.

For this purpose, we present an optoelectronic AFE receiver with an on-chip P^+^/N-well APD realized using low-cost 180 nm CMOS technology, which enables circumvention of the complicated integration issue of an optical device onto an integrated circuit. Moreover, a two-dimensional modified Vernier TDC is employed to generate 4-bit binary codes for the estimation of target distance. Then, a quantized convolutional neural network (CNN) is designed to determine the falling state of patients in an LTCF. Hardware generation and its evaluation are carried out, revealing that the proposed system can fulfil the classification of patient states in real time.

This paper is organized as follows: Section 2 describes the realization of the LiDAR AFE with an on-chip P^+^/N-well APD as an optical detector and provides the measured results. Section 3 describes the realization of the NPU and provides the measured results. Then, conclusions are drawn in Section 4.

## 2. LiDAR AFE

As aforementioned, the LiDAR AFE consists of an on-chip APD for photocurrent generation, a TIA for converting the incoming photocurrents to voltages, a PA for boosting the voltage signals, an output buffer, and a TDC for the estimation of target distance.

### 2.1. Circuit Description

Figure 4a illustrates a cross-sectional view of the P^+^/N-well APD realized in a standard 180 nm CMOS process, where shallow-trench isolation (STI) is exploited as a guard ring to prevent edge breakdown [2]. Figure 4b depicts a schematic diagram of the TIA, in which a voltage-mode CMOS feedforward input configuration is employed to improve the transimpedance gain, i.e., to almost 2× higher than that of a conventional voltage-mode inverter (INV) input stage [2,3].

Figure 4c shows a schematic diagram of the CI-PA, which consists of four inverters and two diode-connected output buffers. Therefore, the output voltages (*V_ON_* and *V_OP_*) can be enhanced by merging the input signals (VCF_N_ and VCF_P_) with other small portions of another path (g_mb_). Provided that the value of g_ma_ is 4 times larger than that of g_mb_, the amplitude and phase mismatches can be considerably reduced [10]. Yet, this CI-PA introduces amplitude mismatches between two outputs in the case of short-distance detection because g_ma_ may vary severely with respect to the variation in v_gs_. Therefore, the circuit design should be carefully conducted to match $\partial g_{man}/\partial v_{gs}\;$ with $\partial g_{map}/\partial v_{gs}$, where *g_man_* and *g_map_* represent the transconductance of NMOS and PMOS at the input inverter stage, respectively.

Post-layout simulations conducted by using the model parameters of a standard 180 nm CMOS process reveal a transimpedance gain of 87.4 dBΩ, a bandwidth of 630 MHz, and a noise current spectral density of 5.69 pA/√Hz. Here, the on-chip APD is modeled as an ideal current source with a parasitic capacitance of 0.5 pF and a current gain corresponding to the responsivity of 2.72 A/W [2].

Figure 4d shows a block diagram of a two-dimensional (2D)-modified Vernier TDC, which converts a narrow pulse to a wide digital signal using a resettable T-latch [12].

Therefore, this 2D-modified Vernier TDC enables alleviation of the timing walk error by steepening up both rising edges of the START and STOP signals. Both the START and STOP signals of the resettable T-latch are delayed by each inverter chain. The T_START signal is delayed by 3 ns, while the T_STOP signal is delayed by 2 ns at each delay line. Hence, the timing resolution of the 2D-modified Vernier TDC can be 1 ns. In this work, the maximum time interval is designed to be 15 ns, with six START delay lines and three STOP delay lines. Correspondingly, this 15 ns time interval is translated to the maximum detection distance of 4.5 m. Meanwhile, timing comparators are implemented using S-R latches, which determine whether the START signal is ahead of the STOP signal at each point in the Vernier plane. Figure 5 depicts an example of the simulated waveforms of the 2D-modified Vernier TDC with a time interval of 10 ns. It can be clearly seen that it generates 15-bit code of 000001111111111, which is converted to 4-bit binary code 1010 via a thermometer-to-binary encoder.

### 2.2. Measured Results

The test chips of the proposed LiDAR AFE IC are fabricated in a standard 180 nm CMOS process. Figure 6 shows the chip microphotograph and its test setup, in which the chip core (APD + TIA + PA + OB) occupies an area of 456 × 153 μm^2^. The DC measurements reveal that the LiDAR AFE IC consumes 51.5 mW from a single 1.8 V supply.

Figure 7 demonstrates the measured frequency response of the proposed LiDAR AFE IC, where a single-ended transimpedance gain (Z_21_) of 95.1 dBΩ and a bandwidth of 608 MHz are measured.

Figure 8a shows the test setup for the optical measurements, in which the proposed LiDAR AFE IC is located on a printed circuit (PC) board with 50 Ω terminations, and an 850 nm laser diode generates 1 ns light pulses at an 80 MHz repetition rate with a 10 mW average power. The distance between the PC board and the laser diode is set to 50 cm.

Figure 8b demonstrates the optically measured pulse responses, where the consecutive light pulses are incident on the on-chip CMOS P^+^/N-well APD with a responsivity of 2.72 A/W, clearly showing differential output pulses. Here, the dark current and the illumination current of the on-chip APD rise sharply at a breakdown voltage of 11.05 V owing to the avalanche multiplication process. With an emitted laser power of 1 mW, the detection range can reach 10 m.

Table 1 compares the performance of the proposed LiDAR AFE IC with other prior arts. In [2], a CMOS P^+^/M-well on-chip APD with a 2.72-A/W responsivity was integrated with the AFE. However, it demonstrated very poorly recovered optical pulses with an 8 mV_pp_ amplitude and a 25 ms pulse width. In [3], a 16-channel off-chip InGaAs PIN-PD array module with a 0.9-A/W responsivity was utilized. Therefore, it exhibited inherent hardware complexity in a multi-channel array configuration and, hence, could not avoid an increase in cost and form factor. Refs. [13,14] exploited off-chip APDs operating at a 905 nm wavelength, which resulted in large power consumption and hardware complexity in an array configuration of multi-channel receivers. On the contrary, this work provides a comparable transimpedance gain and bandwidth performance with a lower noise current spectral density for comparatively low power consumption at little expense of detection range degradation.

Meanwhile, the TDC core occupies an area of 840 × 270 μm^2^ and consumes 20.9 mW from a single 1.8 V supply. Figure 9 demonstrates the measured TDC outputs of the LiDAR AFE IC at (a) 1 ns and (b) 10 ns time intervals. It can be clearly seen that the 4-bit binary code indicates the delay time between the START and STOP signals.

However, it should be noted that this LiDAR sensor obtains information about the target well in the line-of-sight path, but it may be affected by ambient lights or smoke. Moreover, it may not be able to effectively detect patients in a place with partial coverage.

## 3. Neural Processing Unit

To establish a system that determines the fallen or non-fallen status of a patient in the input images from the LiDAR sensor, a quantized convolutional neural network (CNN) is designed. CNN quantization reduces storage and memory requirements, which is important for the deployment of CNN models onto small edge devices. We use the Fallen Person Datasets (FPDS) image dataset [10] as the input.

### 3.1. Quantized CNN Model

Object detection is an image-processing algorithm that locates objects within an image and labels them. YOLO v5 [15] is a state-of-the-art object detection algorithm that can be trained for our fall detection application. However, the YOLO v5 model has tens of millions of parameters, which are not suitable for edge AI devices with only a few megabytes of storage and memory.

Because of the low-precision computation, computation can be significantly reduced compared to the existing IEEE 754 floating point format, thereby increasing the frames per second (FPS) and energy consumption per frame, which are important for targeted edge devices. The structure of the CNN in our work is based on [10], and it is composed of multiple convolution layers, batch normalization layers, connected layers, activation layers, and pooling layers, as shown in Figure 10.

After the final connected layer, a classifier layer determines whether the status is “fallen” or “non-fallen”. The model is trained to detect the existence of a fallen person regardless of other objects in the scene. If there is a fallen person in the frame, it will be detected even if there is another object in the frame, including a non-fallen person. We use Theano [16], a Python library for numerical operations designed to define, optimize, and evaluate mathematical expressions containing operations of multidimensional arrays required to compute large neural networks. In addition, we use Lasagne libraries [17], which support various feedback networks and a high-level application programming interface (API) for easy layer design. The model is trained on the FPDS resized to 32 × 32 pixels and quantized to 2 bits. The classification accuracy after training is approximately 67%.

### 3.2. Design Flow and Toolchain

We implement the quantized CNN on the Python Productivity for Zynq (PYNQ) board [18]. The FINN framework was developed by Xilinx and facilitates hardware generation for data flows and architectures for quantized neural networks. We use BNN-PYNQ [19] based on FINN [20] for the quantization of the trained CNN model. It supports the 1-bit or 2-bit quantization of weights and activations.

The FINN-HLS (high level synthesis) library [21] provided within the framework enables the convenient generation of hardware that can be operated in FPGA through data flow and architectural technology for quantized CNNs.

We use the framework for the efficient development of hardware accelerator design. Hardware generation with HLS refers to high-level synthesis and to the creation of real-world operational hardware through architecture and data flow technologies rather than the traditional hardware design method—register-transfer level (RTL). The advantage of this method is that it can be developed in a higher-level language than the conventional design method, reducing the complexity of development and reducing the time spent. The use of HLS can accelerate design space exploration to find an efficient architecture by reducing the time to evaluate performance with different hardware structures. We use Vivado HLS [22].

### 3.3. Hardware Architecture

As shown in Figure 11, the hardware consists of the Matrix–Vector–Threshold Unit (MVTU) and SIMD structures, where MVTU, a processing element (PE), is the basic unit of operation. The structure of the hardware generated in this task can adjust the throughput of each layer engine by configuring the number of PE and SIMD lanes in the MVTU. The amount of computational logic and the number of SIMD lanes are determined under the constrains of the FPGA used.

Based on the model defined in FINN, the hardware structure is defined as shown in Figure 12a for efficient operation. Each layer has its own dedicated engine, and the amount of calculation per hour can be increased by reducing delay through the streaming structure in which the calculation result of the previous layer begins to be obtained. Considering the hardware resources of Pynq-Z1 board and timing constraint, Table 2 shows the numbers of PEs and SIMD lanes of each layer.

In this design, the parameters for operation may be stored in the on-chip memory for each layer, as shown in Figure 12a, to reduce unnecessary delay by reducing access to separate off-chip memory. Considering the heterogeneous streaming and SIMD structure, the most efficient way to operate the hardware is to set the time required to process each layer similarly so that there is no delay between the layer calculation engines. Each delay can be identified in Figure 12b. We aim to minimize the delay by considering the hardware resources and models. The hardware generation result is presented in Figure 13. Through a performance evaluation, it is confirmed that 2-bit quantization performs best while fitting in the target FPGA.

### 3.4. Evaluation

The FPDS test dataset [10] consists of a total of 973 images, comprising 391 images of people who have fallen and 830 images of people who have not fallen. The demo system used in this work is as shown in Figure 14a. Both the PYNQ-Z1 board and the Host PC are based on Ubuntu and are connected via Ethernet. All commands and files are transferred between the PYNQ-Z1 board and the Host PC via the SSH protocol. The MicroSD card inserted into the PYNQ-Z1 board includes not only the OS image (Ubuntu) for the PYNQ-Z1 board but also the FPDS test dataset image files, the accelerator bitstream to download to the overlay (programmable logic of the PYNQ-Z1 board), and the weight obtained through training. The overall operation of the demo system is as shown in Figure 14.

The Host PC also has information about the FPDS test dataset. Based on information, the Host PC sends a command, including the name of each image file, to the PYNQ-Z1 board in order to perform inference for each test image. After transmission, when the PYNQ-Z1 board completes the inference, the Host PC requests the file, including the inference result, from the PYNQ-Z1 board through SFTP.

The inference result file contains the inferred class and the time spent for the inference. The PYNQ-Z1 board executes the inference code stored inside with the command received from the Host PC. The inference code first uploads the bitstream stored in the MicroSD card to the overlay and reads the image corresponding to the test image file name sent from the Host PC. After upload, neural network acceleration is performed, and when the subsequent execution is completed, the inference result file is saved onto the MicroSD card. The Host PC updates the GUI based on the received inference result file. The GUI is as shown in Figure 14b.

For each test image, the current test progress (Progress), the current test image, its depth image converted using [11], the inference result of the original test image, the ground truth, the prediction result, the accumulated accuracy, and the throughput (FPS) of the accelerator (current processing speed) are displayed. As a result of performing inference on the FPGA with the hardware accelerator, the accuracy is 65.5%, which is similar to the accuracy obtained after training. The precision is 64.8%, and the recall is 40.2%. The inference speed per image is 205 frames per second (FPS) on average. Our experiments using SCALE-SIM [23] show that the proposed system is also more energy efficient than the existing design on conventional processors by multiple orders of magnitude. This is because the size of the CNN is small enough to fit in the on-chip memory, and, therefore, off-chip memory access is not necessary.

## 4. Conclusions

We realized an indoor-monitoring LiDAR sensor that provides a potentially low-cost, low-power solution for detecting the falls of patients with Alzheimer disease. It consists of an efficient optoelectronic AFE circuit with an on-chip P^+^/N-well APD to alleviate the complicated integration issue of an optical device and a 2D-modified Vernier TDC to generate 4-bit binary codes for range detection. To the best of the authors’ knowledge, this is the first attempt to integrate all analog components, including an input optical device, for indoor-monitoring LiDAR sensors. Moreover, a low-latency neural processor is used for image processing. We demonstrated that the proposed system can obtain privacy-preserving depth images of patients and classify them locally without sending them to a server. The proposed neural processing is lightweight enough to be implemented on low-cost processors, yet it is fast enough to perform classification in real time. For practical deployment, it would be important to install the system so that it can cover as much space as possible, excluding space that can potentially cause misclassification, such as beds and sofas. As future work, the proposed system will be further improved to distinguish various situations, such as sleep and cases of falls, non-falls, and falling down, through context awareness.

## Figures and Tables

**Figure 1 sensors-22-07934-f001:**
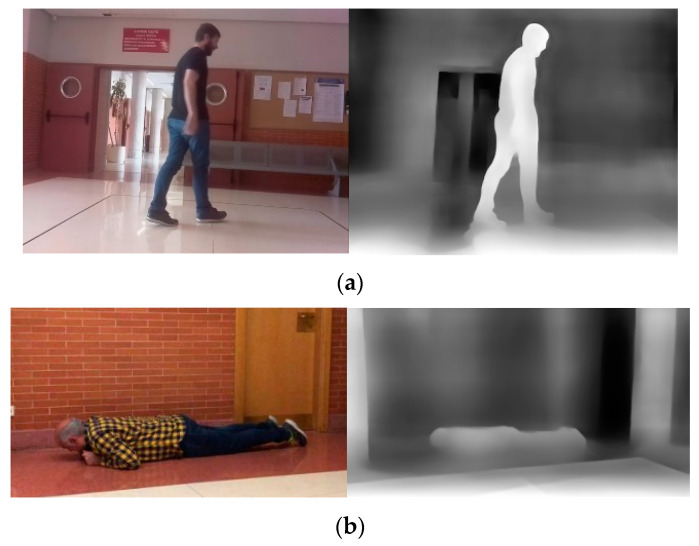
Comparison of RGB and depth images: (**a**) non-fallen and (**b**) fallen (left: RGB images, right: depth images). Images from the Fallen People Dataset (FPDS) [8].

**Figure 2 sensors-22-07934-f002:**
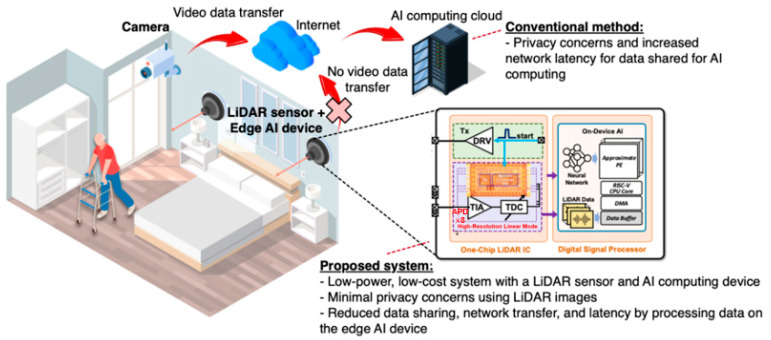
Feasible usage of the proposed indoor-monitoring LiDAR sensor equipped in a living room of LTCF.

**Figure 3 sensors-22-07934-f003:**
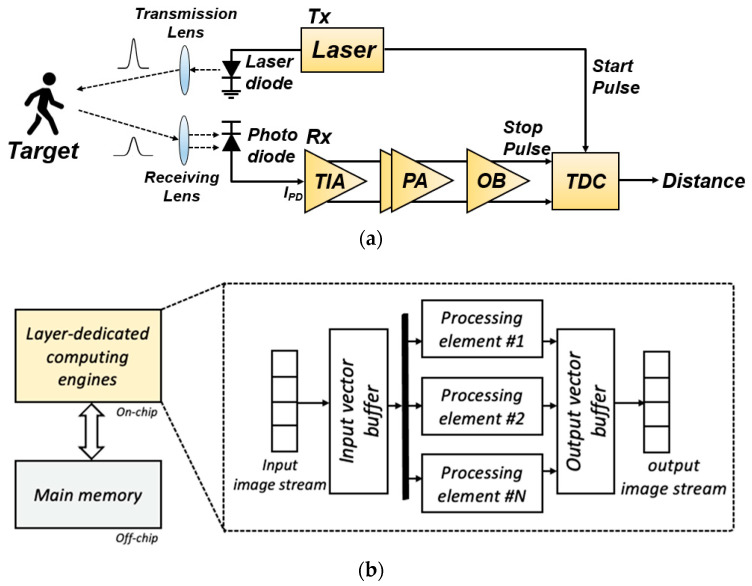
Block diagrams of the proposed optoelectronic Rx IC: (**a**) AFE circuit, (**b**) NPU.

**Figure 4 sensors-22-07934-f004:**
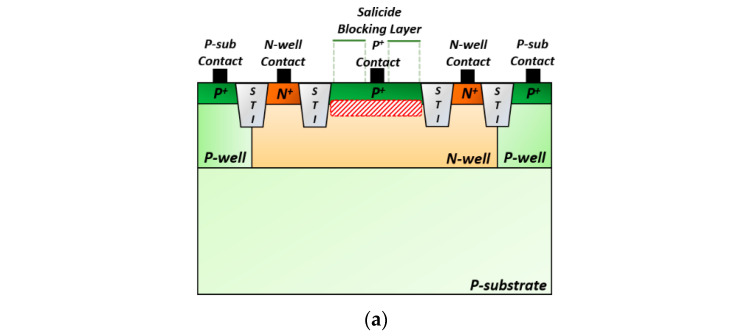
(**a**) On-chip P^+^/N-well APD, (**b**) TIA, (**c**) PA, and (**d**) 2D-modified Vernier TDC.

**Figure 5 sensors-22-07934-f005:**
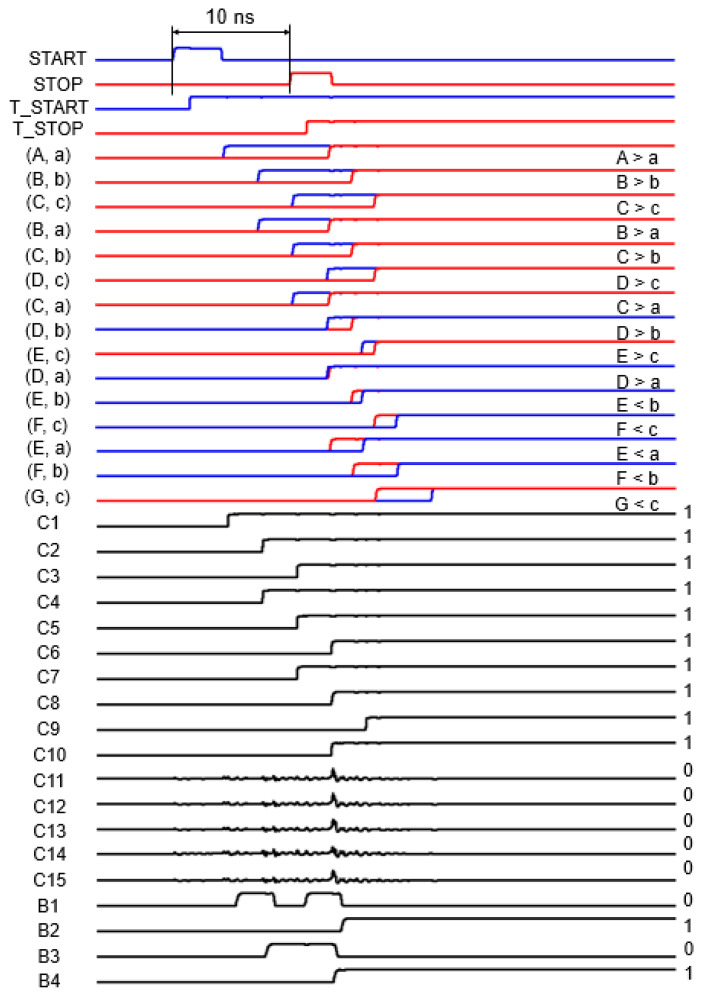
Simulation results of the proposed LiDAR AFE (with a time interval of 10 ns).

**Figure 6 sensors-22-07934-f006:**
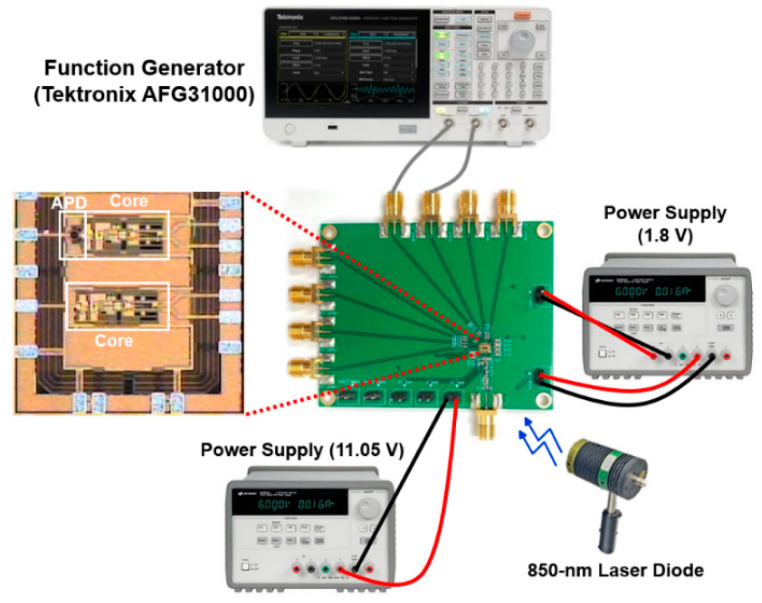
Chip microphotograph of the proposed LiDAR AFE IC and its test setup.

**Figure 7 sensors-22-07934-f007:**
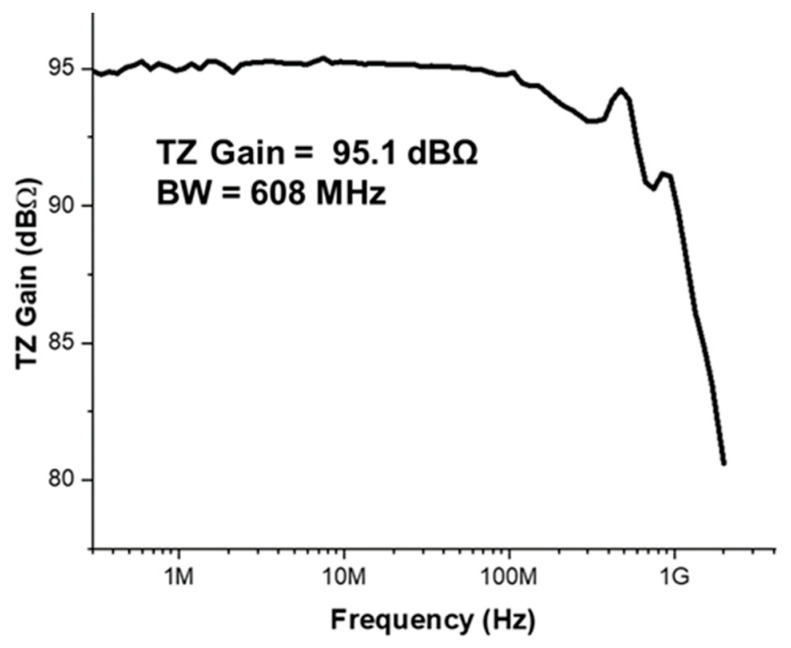
Measured frequency response of the proposed LiDAR AFE IC.

**Figure 8 sensors-22-07934-f008:**
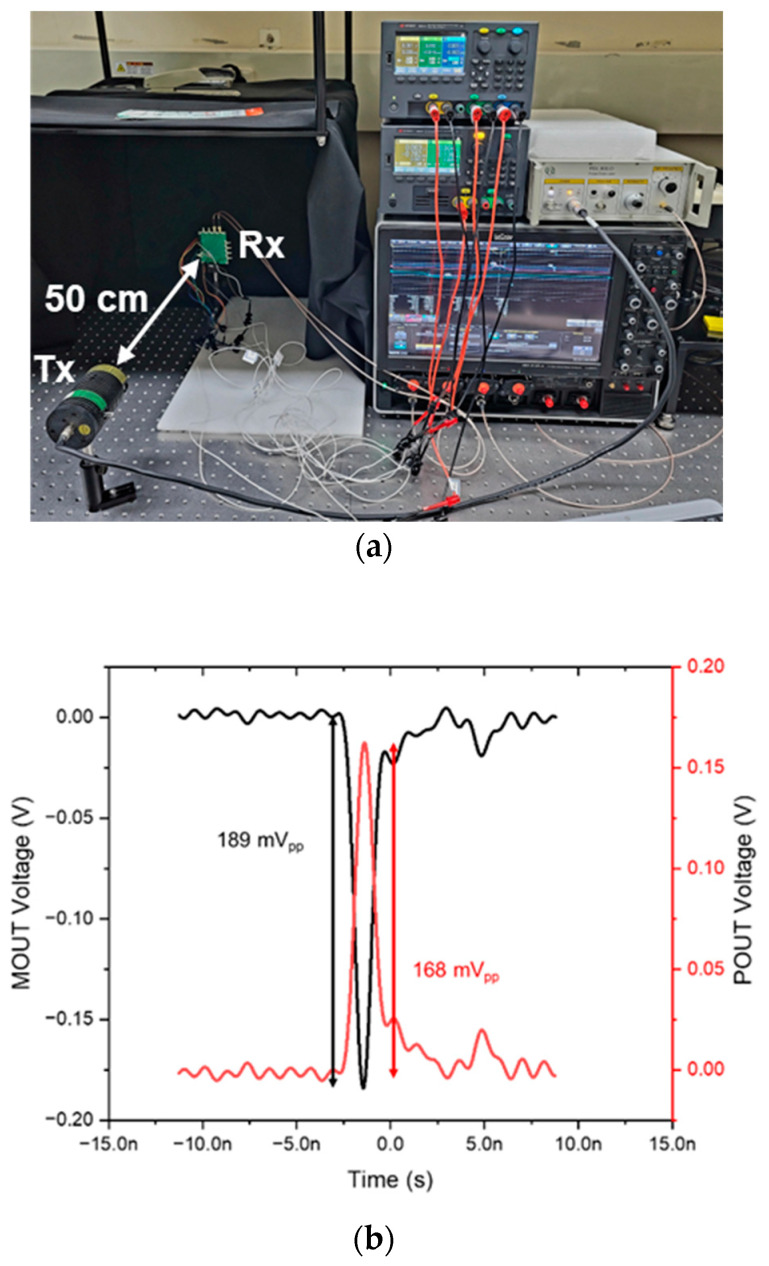
(**a**) Test setup for optical measurements; (**b**) optically measured pulse response.

**Figure 9 sensors-22-07934-f009:**
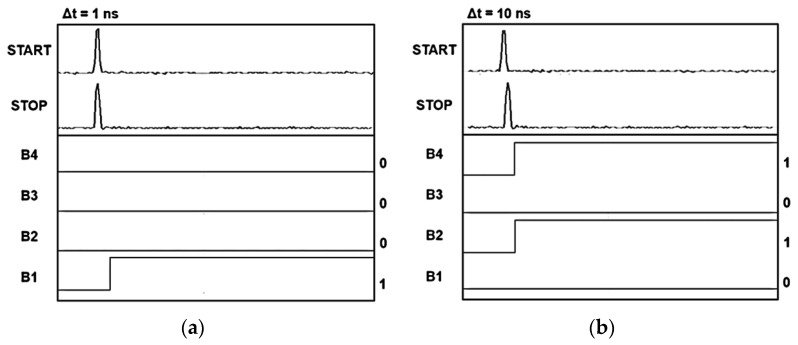
Measured TDC outputs of the LiDAR AFE IC at (**a**) 1 ns and (**b**) 10 ns time intervals.

**Figure 10 sensors-22-07934-f010:**
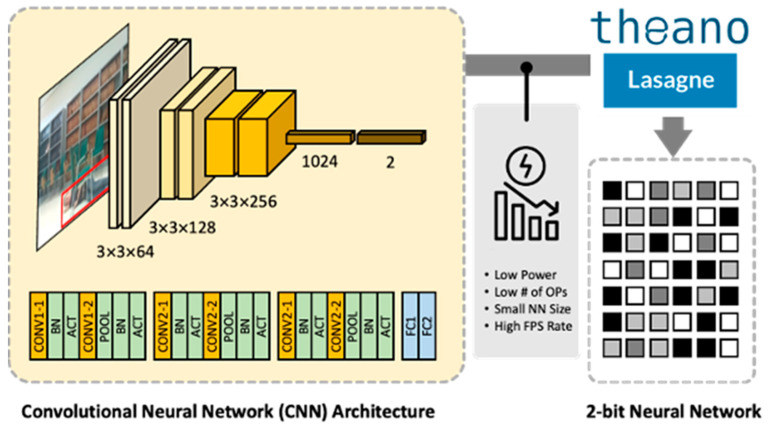
Convolutional neural network architecture.

**Figure 11 sensors-22-07934-f011:**
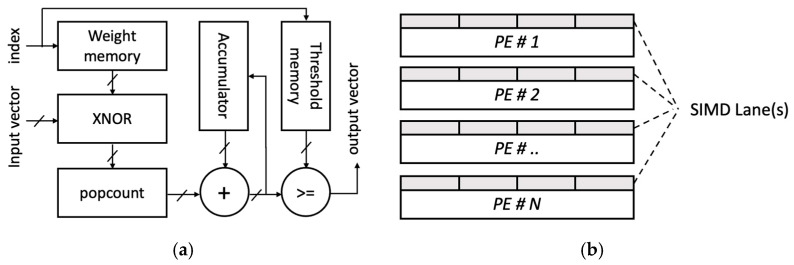
PE and SIMD architecture (**a**) PE architecture, (**b**) SIMD lanes.

**Figure 12 sensors-22-07934-f012:**
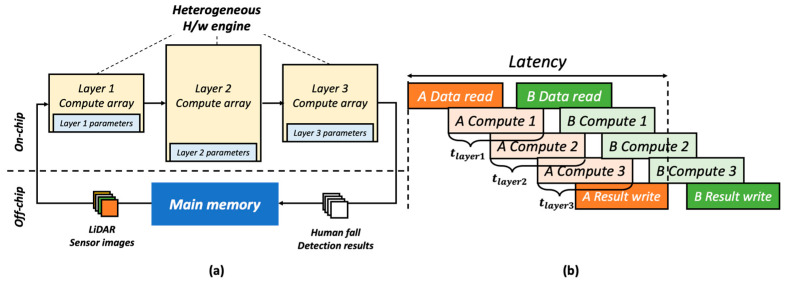
Heterogeneous streaming architecture (**a**) Layer-dedicated H/W engine, (**b**) Pipelined streaming architecture.

**Figure 13 sensors-22-07934-f013:**
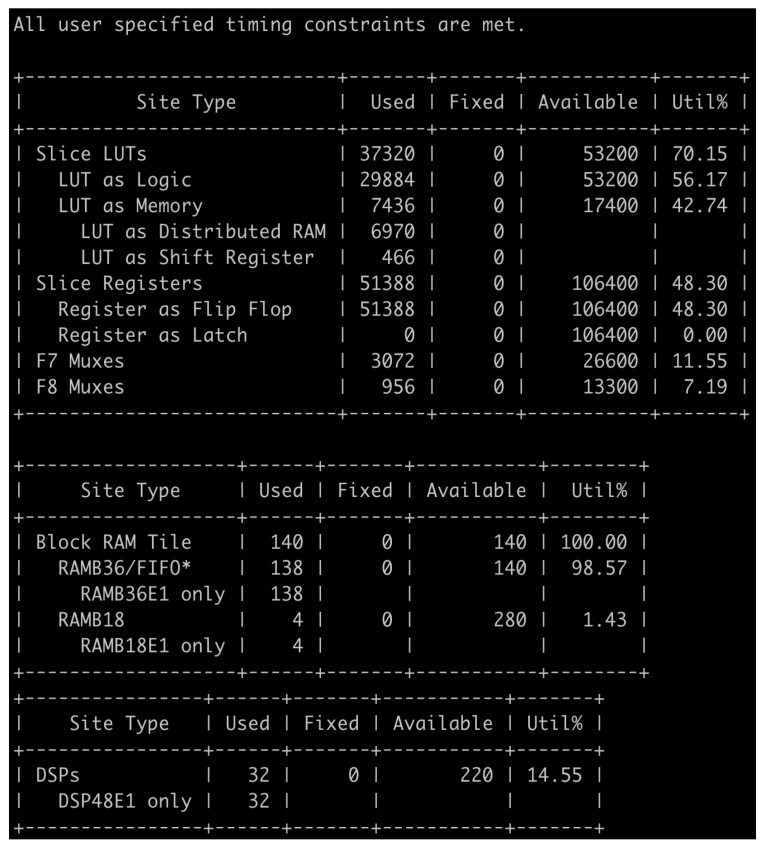
Hardware generation results. * Note: Each Block RAM Tile only has one FIFO logic available and therefore can accommodate only one FIFO36E2 or one FIFO18E2.

**Figure 14 sensors-22-07934-f014:**
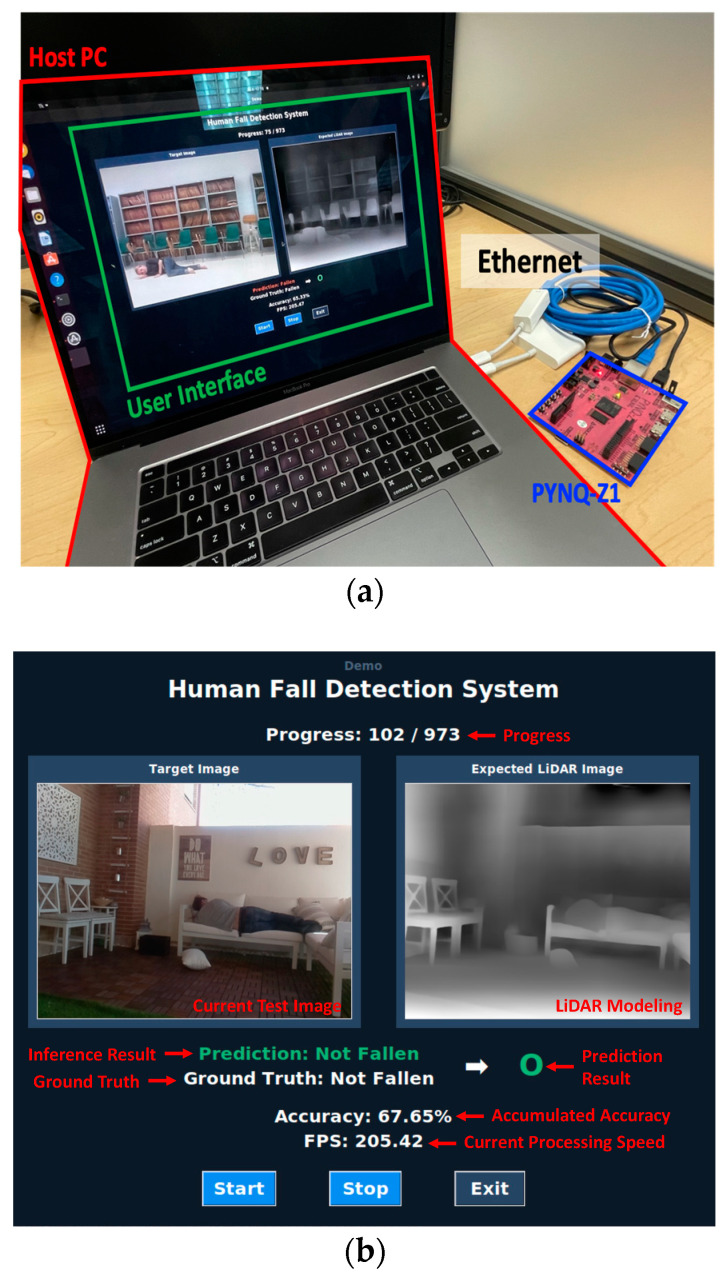
(**a**) User interface of demonstration system and (**b**) demonstration system.

**Table 1 sensors-22-07934-t001:** Performance Comparison of the Proposed LiDAR AFE IC with Other Prior Arts.

Parameters		This Work	[2]	[3]	[10]	[12]
CMOS technology (nm)		180	180	180	350	180
PD	Type	APD (on-chip)	APD (on-chip)	InGaAs PIN-PD (off-chip)	APD (off-chip)	APD (off-chip)
	C_pd_ (pF)	0.5	0.5 *	0.5	3.0	1.2
	Responsivity (A/W)	2.72	2.72	0.9	40	50
	Wavelength (nm)	850	850	1550	905	905
TZ gain (dBΩ)		95.1	93.4	76.3	100	86
Bandwidth (MHz)		608	790	720	230	281
Noise current spectral density (pA/√Hz)		4.54	12	6.3	6.32	4.68
Detection range (m)		10 ^†^	10	25	34	N/A
Power dissipation (mW)		51.5	56.5	29.8	180	200

* Estimated from the measured breakdown voltage. ^†^ Estimated for 1 mW emitter optical power.

**Table 2 sensors-22-07934-t002:** PE and SIMD configuration.

	Conv1	Conv2	Maxpool	Conv3	Conv4	Maxpool	Conv5	Conv6	FC	FC	FC
PE	16	32	-	16	16	-	4	1	1	1	4
SIMD	3	32	-	32	32	-	32	32	4	8	1

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
