# Peer review of "An Indoor-Monitoring LiDAR Sensor for Patients with Alzheimer Disease Residing in Long-Term Care Facilities"

_sensors, 2022, doi:10.3390/s22207934_

Round 1
Reviewer 1 Report
The Alzheimer Patients Residing in Long-Term Care Facilities proposed by the authors is a topic of interest to researchers in related fields, but the paper needs very significant improvement before it can be accepted for publication, and the detailed comments are as follows.
1- It can be seen that the authors have made great contributions and work for smart care, including the independent development of LiDAR and the construction of edge computing network, which solves the problem of privacy leakage in monitoring and provides a new smart care idea for the general scholars.
2- For the innovation point of this work, I found that the authors did not describe it in the article. As a paper with good structure, the authors should give the innovation of this article at the end of the introduction or related work. The authors' work on certain problems. Or, the improvements made by the authors for certain models. Please summarize a few innovations in the manuscript so that the reader can read it with an accurate understanding of the purpose of this work. Instead of reading the whole paper to get the real purpose of your work.
3- For the experimental part of the manuscript, I would like the authors to conduct more experiments to demonstrate the advantages of the proposed algorithm and the designed hardware.
4- In the (CONCLUSION) section the authors should elaborate on the shortcomings of the algorithm in this paper. With all due respect, the conclusion should be rewritten to 1) clearly describe the necessary features/advantages of the proposed method that other methods do not have, and 2) describe the limitations of the proposed method and what aspects of the proposed method can be further improved, why and how.
5- There are several other issues regarding this work as follows.
a) First, it makes sense for the authors to use LIDAR signals for monitoring. However, the reliability of the monitoring system is affected by the fact that the lidar itself is not able to obtain information about the target in the NLOS path and may be affected by light or smoke, so should experiments be designed to demonstrate this work or should the shortcomings of this work be listed in the literature.
b) Second, for monitoring patients. Although privacy is important, we are more concerned about the reliability of monitoring. I think the authors should use some real data to show whether the proposed CNN network can really achieve effective monitoring.
Author Response
[Reviewer 1]
1. It can be seen that the authors have made great contributions and work for smart care, including the independent development of LiDAR and the construction of edge computing network, which solves the problem of privacy leakage in monitoring and provides a new smart care idea for the general scholars.
(ans.) Thanks a lot for this comment.
2. For the innovation point of this work, I found that the authors did not describe it in the article. As a paper with good structure, the authors should give the innovation of this article at the end of the introduction or related work. The authors' work on certain problems. Or the improvements made by the authors for certain models. Please summarize a few innovations in the manuscript so that the reader can read it with an accurate understanding of the purpose of this work. Instead of reading the whole paper to get the real purpose of your work.
(ans.) The summary of innovations is listed at the end of Introduction in the revised manuscript, as below (Page 3, Line 83~90).
“For this purpose, we present an optoelectronic AFE receiver with an on-chip P+/N-well APD realized in a low-cost 180-nm CMOS technology that enables to circumvent the complicated integration issue of an optical device onto an integrated circuit. Also, a 2-dimensional modified Vernier TDC is employed to generate 4-bit binary codes for the estimation of target distance. Then, a quantized convolutional neural network (CNN) is designed to determine the falling state of the patients in LCTF. Hardware generation and its evaluation are demonstrated to reveal that the proposed system can fulfil the classification of the patient states in real-time.”
3. For the experimental part of the manuscript, I would like the authors to conduct more experiments to demonstrate the advantages of the proposed algorithm and the designed hardware.
(ans.) Thank you for this comment. We additionally compared the proposed design to a conventional design that uses an off-chip memory using the SCALE-SIM simulator. The results show that multiple orders-of-magnitude better energy efficiency was measured. We added the following to Section 3.4 (Page 13, Lines 301~305).
“Our experiments using SCALE-SIM [23] show that the proposed system is also more energy-efficient than existing design on conventional processors by multiple orders of magnitude. This is because the size of the CNN is small enough to fit in the on-chip memory, and therefore off-chip memory access is not necessary.”
4. In the (CONCLUSION) section the authors should elaborate on the shortcomings of the algorithm in this paper. With all due respect, the conclusion should be rewritten to 1) clearly describe the necessary features/advantages of the proposed method that other methods do not have, and 2) describe the limitations of the proposed method and what aspects of the proposed method can be further improved, why and how.
(ans.) The conclusion is rewritten in the revised manuscript, as below.
(Page 14, Lines 308~314) “It consists of an efficient optoelectronic AFE circuit with an on-chip P+/N-well APD to alleviate the complicated integration issue of an optical device and a 2D modified Vernier TDC to generate 4-bit binary codes for range detection. To the best of the authors’ knowledge, this is the first attempt to integrate all the analog components including an input optical device for the indoor monitoring LiDAR sensors. Also, a low-latency neural processor is followed for image processing.”
(Page 14, Lines 318~322) “As a future work, the proposed system would be further improved even to distinguish various different situations such as sleep, fall, not fall, fall-down, etc, through context-awareness.”
5. There are several other issues regarding this work as follows.
5a) First, it makes sense for the authors to use LIDAR signals for monitoring. However, the reliability of the monitoring system is affected by the fact that the lidar itself is not able to obtain information about the target in the NLOS path and may be affected by light or smoke, so should experiments be designed to demonstrate this work or should the shortcomings of this work be listed in the literature.
(ans.) Thanks a lot for this comment. This LiDAR sensor obtains the information about the target well in the LOS path and may be affected by ambient lights or smoke. For a place with partial coverage, it may not be able to detect patients effectively. Therefore, it is recommended to install this LiDAR sensor on a ceiling to get a full view, and to install one sensor in each room to avoid the coverage as possible as we can. These shortcomings are included in the revised manuscript (Page 7, Lines 175~177).
5b) Second, for monitoring patients. Although privacy is important, we are more concerned about the reliability of monitoring. I think the authors should use some real data to show whether the proposed CNN network can really achieve effective monitoring.
(ans.) This is a valid concern. We argue that even a system with high reliability of monitoring is of no use if it is not adopted by patients due to privacy concerns. The tradeoff between privacy and utility (in this case, the reliability of monitoring) must be carefully explored and optimized for the maximum benefit of the patient, rather than completely sacrificing one for the other. This has been considered a general design goal of many in-home health monitoring systems, such as WiFi, mmWave, and ultra-wideband (UWB). We address the Reviewer’s point in the Introduction section as follows. (Page 1, Lines 34~39)
“Without addressing the privacy concern, patients will be reluctant to adopt such monitoring systems, which may lead to health-related consequences. To address this problem, researchers have investigated the use of non-traditional imaging technologies such as mmWave [6] and ultra-wideband [7] for privacy-preserving monitoring. Similarly, a solution for privacy-preserving yet accurate AI-based fall detection technique is required.”

Reviewer 2 Report
The work seems interesting and attractive to provide proper health care for elders. However, I have some remarks,
1. Authors need to define RGB in its first appearance, page 1, line 40.
2. Authors need to tell the reader why this paper is unique in the INTRODUCTION section, and how it is different from available research articles.
3. The paper considers fall and not fall cases only, what about other cases? and how I can distinguish between "sleep" and "fall-down"?
4. From Table I, more explanation is required to convince the reader about the importance of this work. for example, this algorithm does not seem to be the best in terms of power consumption, what are the advantages?
5. How will the algorithm behave if there is another human or a pet?
6. For a place with no coverage or partial coverage, how sufficient the algorithm is supposed to be?
7. In 3.4 Evaluation, authors need to make clear the key results of their work, for example, they have 937 images, however, they did not state the successful rate for detecting a "fall". More statistics are required.
8. For someone who wants to apply this algorithm to take care of another person at home, what is required, and how far it is difficult?
Author Response
[Reviewer 2]
The work seems interesting and attractive to provide proper health care for elders. However, I have some remarks.
1. Authors need to define RGB in its first appearance, page 1, line 40.
(ans.) It is defined in the revised manuscript, as below (Page 1, Line 43).
Because LiDAR sensors generate images with depth information but with no detailed RGB (red, green, and blue) information that differentiates individuals, it is considered more suitable for privacy-sensitive applications [6, 7].
2. Authors need to tell the reader why this paper is unique in the INTRODUCTION section, and how it is different from available research articles.
(ans.) The summary of innovations is listed at the end of Introduction in the revised manuscript, as below (Page 3, Lines 83~90).
“For this purpose, we present an optoelectronic AFE receiver with an on-chip P+/N-well APD realized in a low-cost 180-nm CMOS technology that enables to circumvent the complicated integration issue of an optical device onto an integrated circuit. Also, a 2-dimensional modified Vernier TDC is employed to generate 4-bit binary codes for the estimation of target distance. Then, a quantized convolutional neural network (CNN) is designed to determine the falling state of the patients in LCTF. Hardware generation and its evaluation are demonstrated to reveal that the proposed system can fulfil the classification of the patient states in real-time.”
3. The paper considers fall and not fall cases only, what about other cases? and how I can distinguish between "sleep" and "fall-down"?
(ans.) Thanks a lot for this comment. As demonstrated, we have conducted the experiments only for both fall and not fall cases in this paper. Other cases such as sleep and fall-down are not yet investigated since it would be our future research topic to distinguish various different situations. Nevertheless, we believe more sophisticated classification such as “sleeping” versus “fallen” should be achieved though context-awareness. For example, if an object is classified as a fallen person but the location is where a bed or a sofa is previously detected, this is likely a sleeping person. This will require more advanced object detection and classification, which we will explore in the future. We mention this in the Conclusions section as follows.
“As a future work, the proposed system would be further improved even to distinguish various different situations such as sleep, fall, not fall, fall-down, etc, through context-awareness.”
4. From Table I, more explanation is required to convince the reader about the importance of this work. for example, this algorithm does not seem to be the best in terms of power consumption, what are the advantages?
(ans.) Thanks a lot for this comment. As commented, our work is not the best in terms of power consumption. But, the importance of this work lies on (1) the implementation of an on-chip APD in a 180-nm CMOS technology to facilitate the complicated integration issue of an optical device onto the integrated circuits, (2) low-noise performance to enable the optical pulse measurements (which is much better than the case of the reference [2]). Therefore, we revised the sentence in the manuscript, as below (Page 7, Lines 167~170).
“To the contrary, this work provides comparable transimpedance gain and bandwidth performance with lower noise current spectral density for comparatively low power consumption at a little expense of detection range degradation.”
5. How will the algorithm behave if there is another human or a pet?
(ans.) The CNN model is trained to detect the existence of a fallen person, regardless of other objects in the scene. If there is a fallen person in the scene, it will be detected even if there is any other object such as a pet or a non-fallen person in the scene. We clarify this in the manuscript as follows (Page 9, Lines 214~216).
“The model is trained to detect the existence of a fallen person, regardless of other objects in the scene. If there is a fallen person in the frame, it will be detected even if there is any other object in the frame including a non-fallen person”
6. For a place with no coverage or partial coverage, how sufficient the algorithm is supposed to be?
(ans.) Thank you for this comment. Due to the line-of-sight property of LiDAR sensors, the algorithm will not detect objects in such regions, which is desirable for privacy. However, thanks to the potentially low-cost of the proposed system, it can be deployed in scale to maximize the coverage as needed.
7. In 3.4 Evaluation, authors need to make clear the key results of their work, for example, they have 937 images, however, they did not state the successful rate for detecting a "fall". More statistics are required.
(ans.) In Section 3.4, in addition to the success rate (65.5 %) of fall detection, we also provide precision (i.e., TP/(TP+FP)) and recall (i.e., TP/(TP+FN)), where TP is true positive, FP is false positive, and FN is false negative, as follows (Page 12, Line 298~300).
“As a result of performing inference on the FPGA with the hardware accelerator, the accuracy is 65.5%, which is similar to the accuracy obtained after training. The precision is 64.8%, and the recall is 40.2%.”
8. For someone who wants to apply this algorithm to take care of another person at home, what is required, and how far it is difficult?
(ans.) Thank you for this comment. As the Reviewer pointed out in Questions 3 and 6, for the successful deployment, the careful installation of the system is important so as to maximize coverage whiling minimizing false detection of a sleeping person. For example, when installed in a living room, the sensor will need to be installed to cover as much space as possible but not to include the sofa in the frame. We mention it in the Conclusions section as follows. (Page 13, Lines 318~320)
“For practical deployment, it would be important to install the system so in can cover as much space as possible but not to include the space that can potentially cause misclassification, such as beds and sofas.”
